# How do caves breathe: The airflow patterns in karst underground

**Franci Gabrovšek** *

Karst Research Institute, ZRC SAZU, Postojna, Slovenia

* franci.gabrovsek@zrc-sazu.si

## Abstract

Caves and their surrounding fracture systems in the vadose zone of karst regions host a unique atmospheric environment. Understanding the airflow patterns in caves is critical to understanding the properties of the subsurface atmosphere and the chemical interactions between air, water, and rock. The most common driver of airflow in caves is the density difference between the subsurface and the outside air, known as the chimney effect. Observations show that seasonal airflow patterns in caves also depend on the geometry of passages. In this work, I present and use a numerical model of a passage embedded and thermally coupled to a rock mass to study the relationship between the airflow pattern and passage geometry. As the outside air enters the subsurface, it approaches thermal equilibrium with the rock mass along a characteristic relaxation length. This determines the temperature and density contrast between the inside and outside air, and the resulting pressure difference, which drives the airflow. In passages with non-uniform outlines and/ or cross-sections, the relaxation length may depend on the flow direction, resulting in different airflow velocities in cold and warm periods for the same absolute temperature difference between the massif and the external temperature. In a passage with a V-shaped longitudinal profile, the airflow is triggered by instability which causes the feedback between the relaxation length and airflow velocity. The airflow pattern can also be altered by snow and ice. Heat transfer in the rock and the thermal inertia of the rock also change the relaxation lengths and cause hysteresis in the curve presenting the airflow velocity vs. temperature difference.

## Introduction

In karst regions, which account for about 15% of the Earth's ice-free land, solution channels or caves of varying size and complexity are characteristic features of the subsurface [1, 2]. Their development usually begins in the phreatic zone, below the water table, and continues in the vadose zone due to tectonic uplift and/or lowering of the water table [3]. Networks of solution passages and fractures span the entire vadose zone, which can be even more than two kilometres thick [4, 5]. The intersections between the passages and the karst surface represent inlets and outlets of air and water. These can range from large cave entrances to fissures less

**Data Availability Statement:** Data presented in this work are available at: https://cloud.izrk.zrc-sazu.si/index.php/s/s2oTxfqDAADfJJd.

**Funding:** This work was was funded by the Slovenian Research Agency as part of the research projects L6-9397 and J7-4630, and the research

programme P6-0119. The funder had no role in study design, data collection and analysis, decision to publish, or preparation of the manuscript.

**Competing interests:** The author has declared that no competing interests exist.

than a centimetre wide. Water may enter the vadose zone through the infiltration of precipitation or concentrated via sinking streams (Fig 1a).

Airflow in caves may be driven by several mechanisms: the most common is the chimney effect [6], in which the difference in density between the subsurface air and the outside air results in pressure differences that drive the subsurface airflow. The subsurface airflow may also be driven by other mechanisms, such as barometric variations in the external atmosphere [7] or dynamic pressure effect caused by external winds [8].

Understanding the airflow patterns in karst massifs has several important implications. In carbonate karst, the solubility of calcite and dolomite depends on the availability of $CO_2$ in the (ground) water [9]. The vadose zone represents an open system in terms of the dissolution of carbonates. This means that the water is in contact with the rock and air and that $CO_2$ consumed for carbonate dissolution is replenished from the atmosphere. The $CO_2$ content in the atmosphere is therefore a critical factor in the evolution of vadose caves. $CO_2$ concentration in the vadose zone also controls the deposition of speleothems, which have become an important paleoclimate proxy [10, 11]. The spatial and temporal distribution of $CO_2$ is controlled by the distribution of its sources and sinks and by transport mechanisms, with airflow being the most important [12–14]. Understanding the mechanisms that drive airflow in the karst vadose zone is therefore important for understanding the distribution of $CO_2$ in karst, the development of caves in the vadose and epiphreatic zone, the deposition of speleothems, and the role of karst processes in the global carbon budget [15].

Physical exploration and survey of caves is often the only source of information on the structure of the vadose zone. During cave explorations, cavers observe and track airflow, which is an important indicator of possible cave continuation. This is especially important at constrictions or breakdowns, where it is not possible to see beyond the obstacle.

The work is organised as follows:

- Basic concepts of the chimney effect are reviewed and confronted with airflow observations in real systems,

- a numerical model is presented, which couples a density-driven airflow in a simple pipe and heat exchange between the air and the rock,

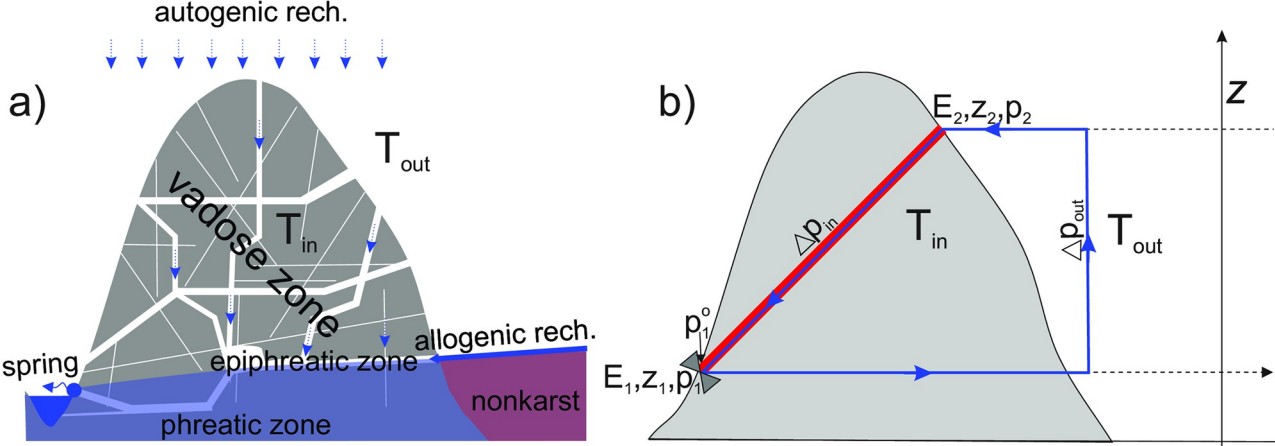

**Fig 1.** a) An idealised cross-section through a karst massif presenting the main hydrological zones, recharge modes and conduit/fracture system in the vadose zone. b) A single channel extending between two entrances $E_1$ and $E_2$ at elevations $z_1$ and $z_2$. $T_{in}$ and $T_{out}$ are the temperature of the massif and the outside temperature. The sum of the pressure changes along the blue line gives the driving pressure for the ventilation (see Eqs 3 to 6).

- the model is used to demonstrate the role of passage longitudinal profile, cross-section and other potential factors, which control the airflow patterns,

- to discuss the modelling results in-depth, an analytical approximation for driving pressure in different situations are given.

## A chimney effect: Basic concepts

The chimney effect in caves is a well-known concept, described in classical textbooks on speleology [6] and cave climate [16, 17]. The density difference between the outside air and the cave air controls the driving pressure of the chimney effect. Besides temperature, water vapour and $CO_2$ are the components that can cause air density variations in karst caves. The $p_{CO_2}$ in caves can reach several per cent, but in ventilated passages it rarely exceeds 1%. Except for the entrance part, caves are humid environments. In temperate climates, relative humidity in caves is generally close to 100%. In this work air density is calculated from the equation compiled by Picard et al. [18]:

$$\rho[\text{kg/m}^3] = \frac{pM_a}{ZRT}\left(1 - x_v(1 - M_v/M_a)\right) \qquad (1)$$

$$M_a = [28.96546 + 12.011 \cdot (x_{CO_2} - 0.0004)] \cdot 10^{-3}\text{kgmol}^{-1} \qquad (2)$$

where $p$[Pa] is the pressure, $R$ is the gas constant, $T$[K] is the thermodynamic temperature, $Z$ is a compressibility factor, here taken as $Z \approx 1$, $M_a$ and $M_v$ are the molar mass of dry air and the molar mass of water, respectively, and $x_v$ and $x_{CO_2}$ are the mole fractions of water and $CO_2$.

Fig 2 shows the variation of air density between -12°C and 25°C for dry air with $p_{CO_2} = 0.01$ atm and air with $RH = 100\%$ and $p_{CO_2} = 0.0004$ atm. To account for the influence of $CO_2$ content and humidity variations, several authors [19] suggest using the virtual temperature $T_v$, which is the temperature of dry air without $CO_2$ with the same density as humid air with some content of $CO_2$ and temperature $T$; $\rho(T_v, x_w = 0, x_{CO_2} = 0) = \rho(T, x_w, x_{CO_2})$. Compared to the real temperature, the virtual temperature decreases with the content of $CO_2$ and increases with RH. The inset in Fig 2 shows the concept; the arrowed dotted line shows the difference between the real temperature $T$ and the virtual temperature $T_v$ at 1% $CO_2$.

In temperate climates with an annual temperature amplitude exceeding 20°C, temperature variations are the main cause of density variation. Therefore, in this work I consider the temperature differences as the main driving factor of the subsurface airflow.

### Driving pressure of the chimney effect: An isothermal approximation

To get a first idea of the driving pressure of the chimney effect, we take air as an ideal gas and its density as a pure function of temperature. For now, we assume that both, the cave temperature $T_{in}$ and the external temperature $T_{out}$ are uniform. Fig 1b shows a single passage with two inputs at elevations $z_1$ and $z_2$. For better understanding, imagine a valve at input $E_1$. The pressure difference between the two sides of the valve is equal to the sum (integral) of the pressure changes along the closed loop that includes the channel and the outside atmosphere, represented by the blue line in Fig 1b. Starting from the inlet $E_1$ at the elevation $z_1$ and pressure $p_1$, following the outer path to $E_2$ at $z_2$ and $p_2$ and along the cave back to $z_1$ and pressure $p_1^o$,

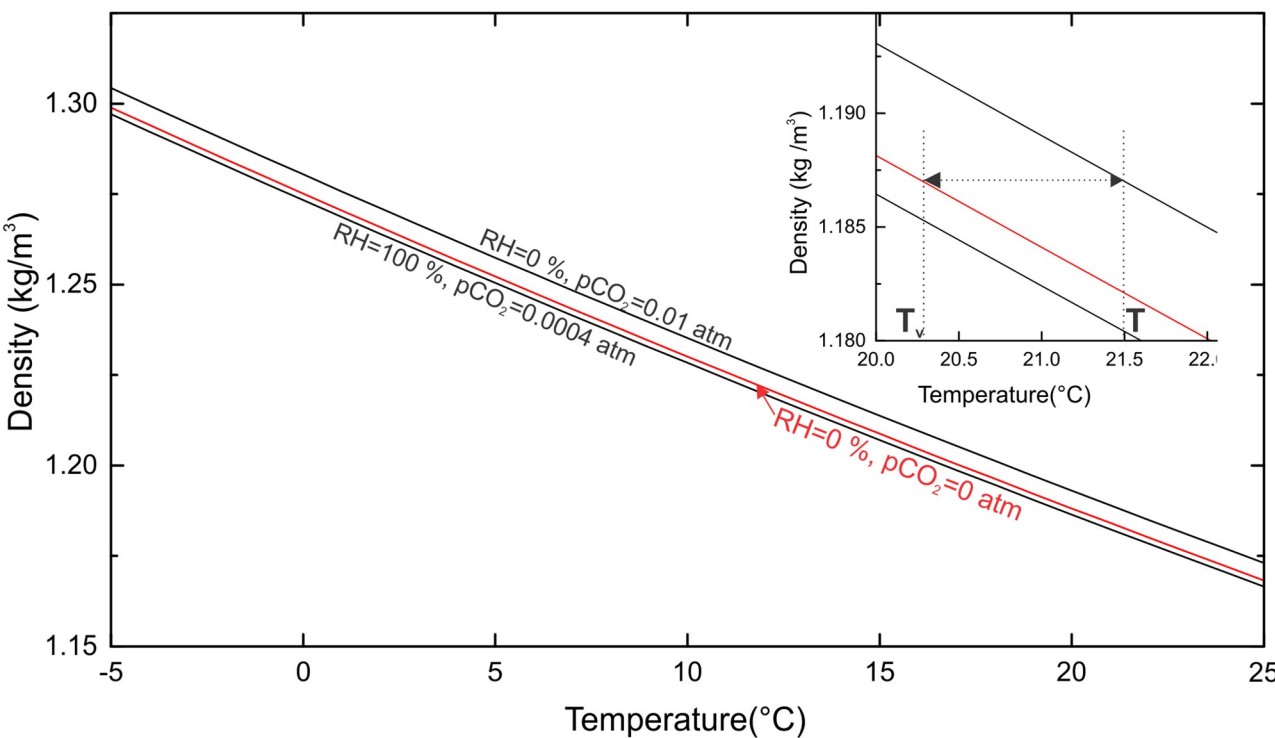

**Fig 2. Variation of air density with temperature for different values of CO$_2$ and relative humidity (RH).** Insert shows the concept of the virtual temperature: air with $T$ = 21.5°C, $RH$ = 0 and $p_{CO_2}$ = 0.01 atm has a virtual temperature $T_v$ = 20.2°C.

we get:

$$p_2 = p_1 e^{-\frac{gM}{RT_{out}}(z_2 - z_1)} \tag{3}$$

$$p_1^o = p_2 e^{\frac{gM}{RT_{in}}(z_2 - z_1)} = p_1 e^{-\frac{gM}{RT_{out}}(z_2 - z_1)} e^{\frac{gM}{RT_{in}}(z_2 - z_1)} \tag{4}$$

$$\Delta p = p_1 - p_1^o = p_1 \left( 1 - e^{-\frac{gM}{RT_{in}T_{out}}(T_{in} - T_{out})\Delta z} \right), \tag{5}$$

where $g$ is the gravitational acceleration and $M$ is the molar mass of air. For small exponents Eq 5 becomes

$$\Delta p = p_1 \frac{gM\Delta z \Delta T}{RT_{in}T_{out}}, \tag{6}$$

where $\Delta T = T_{in} - T_{out}$. The approximation in Eq 6 is the difference between the static pressures of the isothermal incompressible columns of the outside and the cave air. The driving pressure given by Eq 6 is proportional to the temperature difference between the cave and the outside atmosphere and the elevation difference between the two inputs. Removing the fictitious "valve", the airflow is released such that the frictional forces compensate for the driving pressure. Note that $\Delta p > 0$ for $\Delta T > 0$ and $\Delta p < 0$ for $\Delta T < 0$, resulting in airflow from the lower to the upper entrance (*updraft*) during cold periods and from the upper to the lower entrance (*downdraft*) during warm periods. Due to the nonlinearity of Eq 5 the updraft driving pressure in the cold season is slightly higher than the downdraft driving pressure in the warm season for the same $|\Delta T|$ [20]. Typical values of $\Delta p$ are in the order of 50 Pa for $\Delta h$ = 100 m and

$\Delta T = 10°C$. For a turbulent flow, the airflow velocity is proportional to the square root of the driving pressure $\Delta p$. Therefore, $v \propto \sqrt{\Delta T \Delta z}$. Such a square-root relationship has been observed in many caves [20, 21]. The above approximation explains the general relationships between altitude differences, temperature differences, and subsurface airflow.

## Real world scenarios: An example from Postojna Cave, Slovenia

A close look at observations of airflow in real cave systems raises questions that cannot be answered by the approximation given above. An example is given on Fig 3. It shows the context and the results of airflow measurements in Postojna Cave, Slovenia (Fig 3a and 3d). The cave system is over 25 km long and has complex and multiple microclimatic patterns. Here I present the case of two dead-end passages that deviate less than 100 m apart from the main cave passage (Fig 3a). In both passages, the airflow velocity was continuously observed near the intersection with the main passage (Fig 3a). The graphs in Fig 3e and 3f show $|v|(T_{out})$ for 2-year-long record of airflow in both passages. The black line is a square-root fit of the point clouds of measured values. The graphs are clearly asymmetric, with PP showing higher ventilation during warm periods and BP showing higher ventilation durng cold periods. As the entrances from the main passage to both passages are very close, the difference can only be due to differences in their characteristics and their connection to the surface as shown in Fig 3b

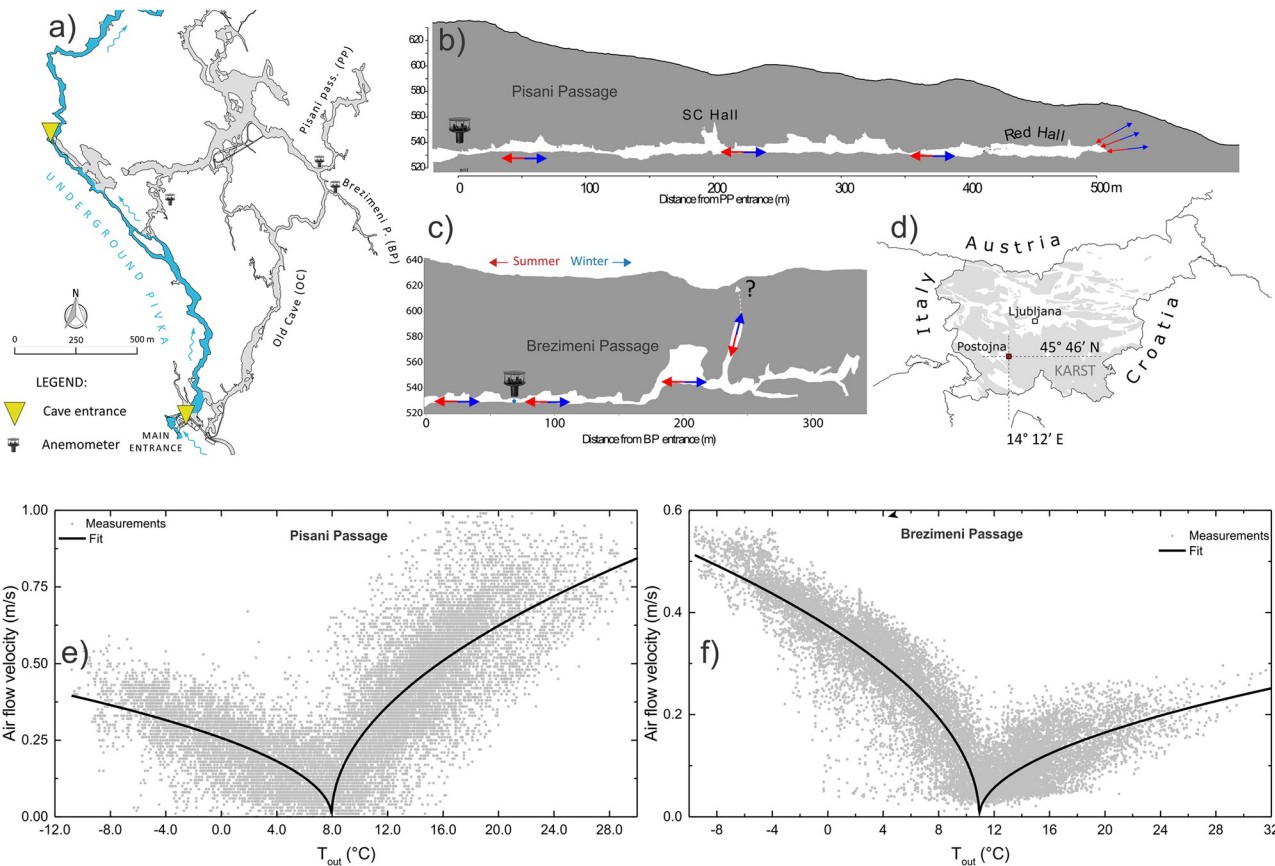

**Fig 3. Map of the Postojna cave.** (a) and simplified profiles of Pisani Passage (b) and Brezimeni Passage (c). Arrows in profiles indicate the direction of airflow in winter (blue arrows) and summer (red arrows) d) Geographic position of Postojna. e) Relation between airflow velocity in Pisani Passage and outside temperature. f) Relation between airflow velocity in Brezimeni Passage and outside temperature. Grey dots present a point cloud of two years of measurements and the black line is the square root fit. Figures a, b, c and d adopted from [14].

and 3c. In BP the airflow to the surface follows a dominant chimney. The chimney does not open to the surface, but a high inflow of warm surface air with low $CO_2$ in summer indicates clearly a strong aeraulic connection to the surface. The airflow path through BP has a distinct L-shaped profile. At PP, the airflow from the passage to the surface does not follow a single dominant path but is distributed among a system of fractures and small channels. In addition, the thickness of the roof above the passage is small, so the profile of the airflow pathway is not a distinct L-shape. Airflow measurements in Postojna cave were part of a broader project, where other cave climate parameters were observed as well. A detailed description of the site and observations are given in [8, 14].

We have been observing seasonal airflow asymmetry in two other observed caves, however, a detailed study of these sites is still in progress, and therefore not presented here. In the following sections, I begin with a conceptual model and continue with a numerical model and analytical approximation to discuss the role of thermal relaxation length in airflow patterns.

## The role of thermal relaxation length

The air temperature in the cave is not constant, but it takes some relaxation length (also thermal length scale, penetration length [22, 23]) for the outside air to reach thermal equilibrium with the massif. If we assume constant wall temperature and convective heat exchange between air and rock, the temperature as a function of distance from the entrance is given by [22, 23]:

$$T(x) = T_{in}(1 - \delta T e^{-x/\lambda}), \tag{7}$$

where $\delta T = (T_{in} - T_{out})/T_{in} = \Delta T/T_{in}$ and $\lambda$ the relaxation length. This depends on the thermodynamic properties of the fluid and on the velocity and hydraulic diameter of a passage. Wiggley and Braun [23] give a relation for $\lambda$ in a tube based on the Dittus-Boelter equation, which relates Nusselt number to Prandtl and Reynolds numbers:

$$\lambda = 15D^{1.2}v^{0.2} \tag{8}$$

Values of $\lambda$ and $D$ are in m, and $v$ in m/s. Note a strong dependence of $\lambda$ on diameter. Weak dependence on the velocity is due to the fact that with increasing velocity an air parcel moves faster along the passage, giving it less time for heat exchange; but at the same time the thermal boundary layer is thinner and the heat transfer between air and rock is more effective. For a passage with a diameter of 2.5 m and airflow velocity of 0.5 m/s, the value of $\lambda$ would be approximately 42 m. Non-zero relaxation length also influences airflow. The longer the relaxation length, the lower the contrast between the cave and outside temperature and density (Fig 4a). The resulting driving pressure is generally lower than predicted by Eq 3. Due to non-zero relaxation length, we also anticipate seasonal asymmetry of airflow patterns in passages with non-uniform longitudinal profiles or cross-sections. As an example consider a passage with an L-shaped outline (Fig 4b). The pressure difference builds up only along the vertical part of the passage. During updraft, the air first flows along the horizontal part, where it approaches thermal equilibrium with the massif. Therefore, the air is close to or in thermal equilibrium with the massif along the entire vertical part, where the pressure builds up, resulting in (near) maximum driving pressure. During downdraft, the warm air first thermally equilibrates along the vertical part, resulting in a lower density contrast with the outside air and a lower driving pressure and airflow velocity than during updraft for the same $|\Delta T|$. The opposite is valid if the passage is horizontal at the level of the higher entrance. Similarly, if the passage diameter is smaller at one entrance and large at the other entrance, the relaxation length will be shorter and the driving pressure and airflow velocity will be higher when the direction of airflow is such that it first enters the segment of small diameter; if the lower part of the passage is small,

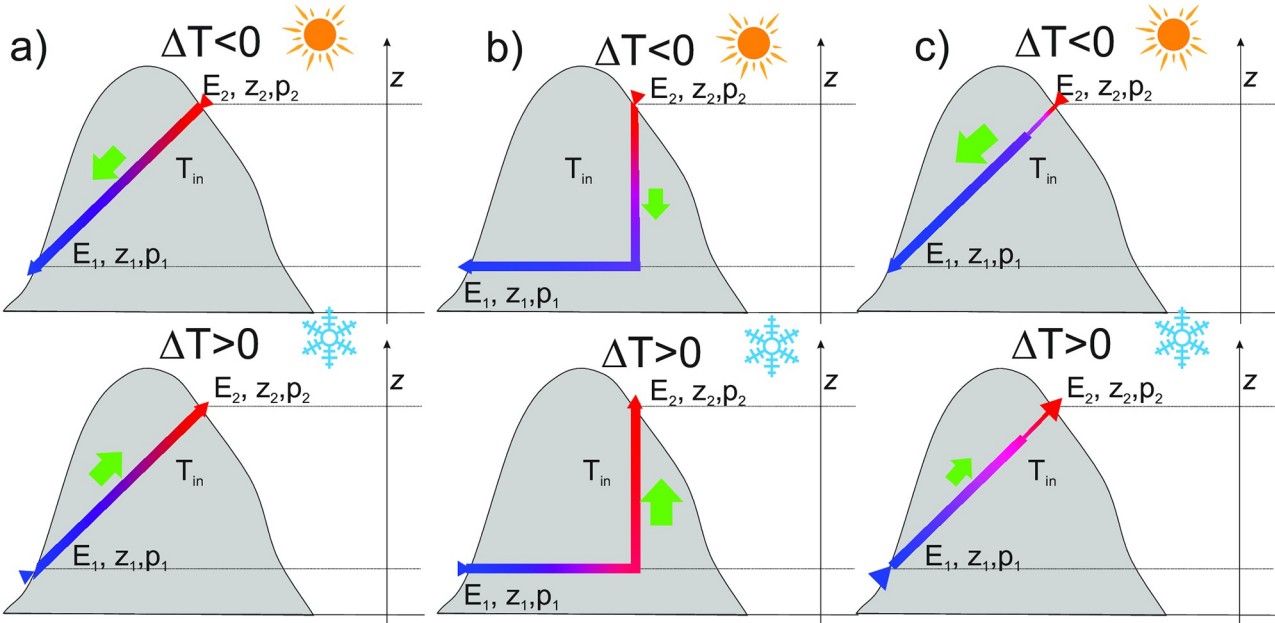

**Fig 4.** Anticipated airflow direction and magnitude in warm (top) and cold (bottom) periods for different settings. a) Uniform slope and cross-section, b) L-shaped outline, c) Passage with a smaller diameter in the upper section.

the ventilation will be stronger in cold period and vice versa, small channels close to the upper entrance promote stronger summer ventilation (Fig 4c). This is also shown schematically in Fig 4. The reasoning explains the presented field cases, with winter dominant ventilation in BP (L-shape) and summer dominant ventilation in PP (small pathways between the passage and surface above it).

The relation between airflow, passage geometry and heat exchange between air and rock is therefore important for understanding airflow patterns. Furthermore, the conduction of heat in the rock cannot be neglected. To account for both, convection and conduction, I now present and use a numerical model which allows studying the role of heat conduction in the surrounding rock massif.

## Methods

### Modelling chimney effect in a single passage

The results of this work are based on a numerical model that couples the density-driven airflow through a circular passage with the heat exchange between the air and the rock massif. The model geometry is shown in Fig 5. It consists of a pipe embedded in and thermally coupled to the surrounding rock mass. The pipe has two entrances at different elevations and is divided into two sections whose length, slope or cross-section may differ.

The model workflow is composed of the following sequence of tasks:

1. Initiate the system and set the boundary conditions (see Table 1) at time $t = 0$,

2. Calculate driving pressure,

3. Calculate airflow velocity/rate in the tube,

4. Calculate advective heat transfer, heat exchange with the rock, heat transfer within the rock, air temperature and density along the tube,

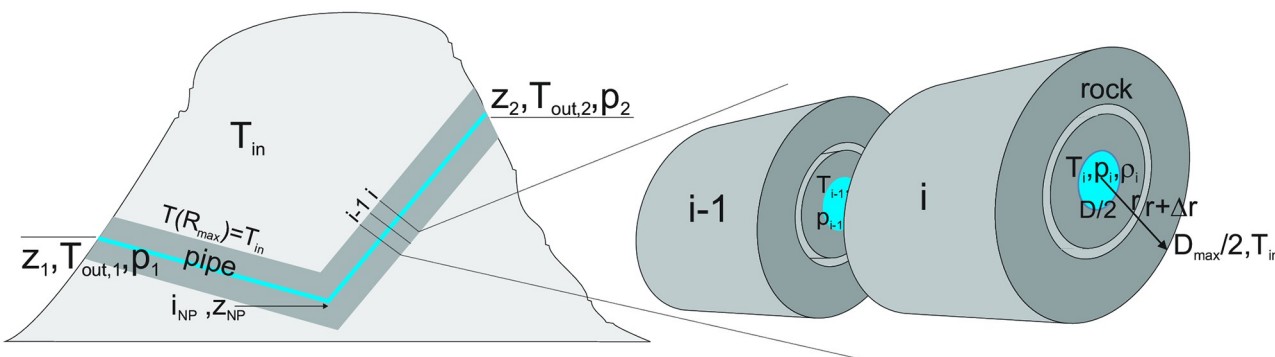

**Fig 5.** Left: Geometry and parameters of the system. The passage is shown by the blue line. Right: Excerpt with two pipe segments and the surrounding cylindrical rock mass.

5. Increase time for $\Delta t$

6. Return to Step 2 or Exit according to defined criteria, such as maximal time.

**Input parameters.** The system is determined by the variations of the external temperature $T_{out}(t)$ and a set of parameters that define its geometry, the thermal properties of the rock and air, and the aeraulic properties of the airflow in pipes. A list of parameters with characteristic values is given in Table 1.

**Calculation of the driving pressure.** The driving pressure is calculated from the difference between the change in external pressure between $z_1$ and $z_2$ and the corresponding change in pressure along the passage. The change in external pressure is calculated from the standard atmosphere equation with the temperature lapse rate $\Gamma[K/km]$. The temperature $T_1$ at the entrance $E_1$ and elevation $z_1$ is given as an input parameter. The pressure at $E_2$, $z_2$ is given by:

$$p_2 = p_1 \left( \frac{T_1 - \Gamma z_2}{T_1 - \Gamma z_1} \right)^{\frac{gM}{RT}} \tag{9}$$

**Table 1. Model parameters.**

| Parameter name | Label | Value / Unit |
|---|---|---|
| **Geometry of the system** | | |
| Elevation of entrances | $z_1, z_2$ | [m] |
| Tube diameter | $D$ | [m] |
| Tube length | $L$ | [m] |
| Elevation and position of knickpoint | $z_{NP}, i_{NP}$ | [m], [] |
| **Other Parameters** | | |
| Temperature, pressure at $E_1$ | $T_1, p_1$ | [K], [Pa] |
| Wall roughness | $\epsilon$ | [], D/100 |
| Massif temperature | $T_{in}$ | [˚C] |
| Rock heat diffusivity | $\alpha$ | $1.14 \cdot 10^{-6} m^2 s^{-1}$ |
| Rock heat conductivity | $\kappa_{rock}$ | 1.3 W m$^{-1}$ K$^{-1}$ |
| Air heat conductivity | $\kappa_{air}$ | $2.2358 \cdot 10^{-4} \cdot T[K]^{0.8535}$ W m$^{-1}$ K$^{-1}$ |
| Heat convection coefficient | $h$ | W m$^{-2}$K$^{-1}$ |

The pressure variation along the pipe is calculated by finite differences. The passage is divided into N segments of length $\delta l_i = L/N$ The elevation change within the i-th segment is given by (see Fig 5).

$$\delta z_i = \begin{cases} (z_{NP} - z_1)/(i_{NP} \cdot \delta l_i), & i < N_m \\ (z_2 - z_{NP})/((i - i_{NP}) \cdot \delta l_i), & i >= N_m \end{cases} \qquad (10)$$

The total change of internal pressure is a sum of pressure changes along all segments:

$$\Delta p_{in} = g \sum_{i=0}^{i=N-1} \rho_i(T_i, pCO_{2,i}, W_i...) \cdot \delta z_i \qquad (11)$$

The density $\rho_i(T_i, pCO_{2,\ i}...)$ in a segment is calculated from Eqs 1 and 2.

$$\rho_i = \frac{p_i \cdot M_{air,i}}{R \cdot T_i}(1 - W*(1 - M_{air}/M_{water})), \qquad (12)$$

where the pressure $p_i$ at i-th segment is calculated from iteration:

$$\Delta p_i = p_{i-1} - g \cdot \rho_{i-1} \cdot \Delta h_{i-1}, \quad p_0 = p_1. \qquad (13)$$

Finally, the driving pressure is given by $\Delta p = \Delta p_{out} - \Delta p_{in}$.

**Calculation of the airflow velocity.** Once the driving pressure is known, the flow velocity is calculated from an explicit approximation of the Colebrook-White equation [24]:

$$v = -0.965\sqrt{\frac{D^5\Delta p}{L\rho_{av}}}\ln\left[\frac{\epsilon}{3.7D} + \sqrt{\frac{3.17v^2L\rho_{av}}{D^3\Delta p}}\right], \qquad (14)$$

where $D$ is the effective diameter of the pipe. The parameters of the equation are given in Table 1. The viscosity of the air is calculated from the Sutherland equation [25].

**Calculation of heat transfer.** The advective heat transport along the tube with heat exchange at the wall is described by:

$$\frac{\partial T}{\partial t} = -v\frac{\partial T}{\partial x} + \frac{4h}{\rho c_p D}(T_{rock}(D/2, x) - T(x)) - \Gamma(dh/dx)dx$$
$$T(x = 0, x = L) = T_1, T_2 \qquad (15)$$

The first term on the right side represents the advective heat transport, the second term represents heat exchange with rock at the wall and the third adiabatic lapse rate due to auto(de) compression. The solution is obtained by explicit, second-order accurate in time and space, Lax-Wendroff scheme [26]. Each pipe element is thermally coupled to a rock mass. The rock conducts heat from or towards the pipe walls and cools or heats the air. A constant temperature of the mass $T_{in}$ is assumed for $r >= R_{max}$ and convective boundary conditions at $r = D/2$. The temperature field in the rock and at the wall is calculated from the heat transfer equation and boundary conditions. Explicit finite differences in cylindrical coordinates are used for the

solution.

$$\frac{\partial T_{rock}}{\partial t} = \alpha \left[ \frac{1}{r} \frac{\partial T_{rock}}{\partial r} + \frac{\partial^2 T_{rock}}{\partial r^2} + \frac{\partial^2 T_{rock}}{\partial x^2} \right]$$

$$T_{rock}(r = D/2) = T_{in}, T_{rock}(x = 0, x = L) = T_1, T_2 \tag{16}$$

$$\kappa_{rock} \left. \frac{\partial T_{rock}}{\partial r} \right|_{r=D_{max}/2} = -h(T_{rock}(D/2, x) - T(x))$$

The convection coefficient $h$ is related to the Nusselt number, the latter being calculated from the Dittus-Boelter relation assuming $Pr = 0.7$ [23],

$$h = \frac{Nu \cdot \kappa_{air}}{D} \tag{17}$$

$$Nu = 0.021 Pr^{0.6} Re^{0.8}. \tag{18}$$

To ensure the stability of the Lax-Wendroff scheme, the time step must be shorter than half the minimum flow through time in any of the pipe segments, $\Delta t_{LW} < 0.5 \times \min(l_i/v_i)$.

## Results and discussion

### Basic scenario

Fig 6 presents the results of the scenario of a 1 km long straight tube (Fig 1b) with a diameter of 2 m and $\Delta h = 100$ m, later referred to as a standard case. The outside temperature is constant at -5˚C, and the massif temperature $T_{in} = 10$˚C. Curves show the airflow velocity, driving pressure, relaxation length, and temperatures at different locations along the tube for 100 days. Fig 7 shows the air temperature in the tube and the rock temperature within the 5 m radius of the surrounding massif at 20, 40 and 80 days.

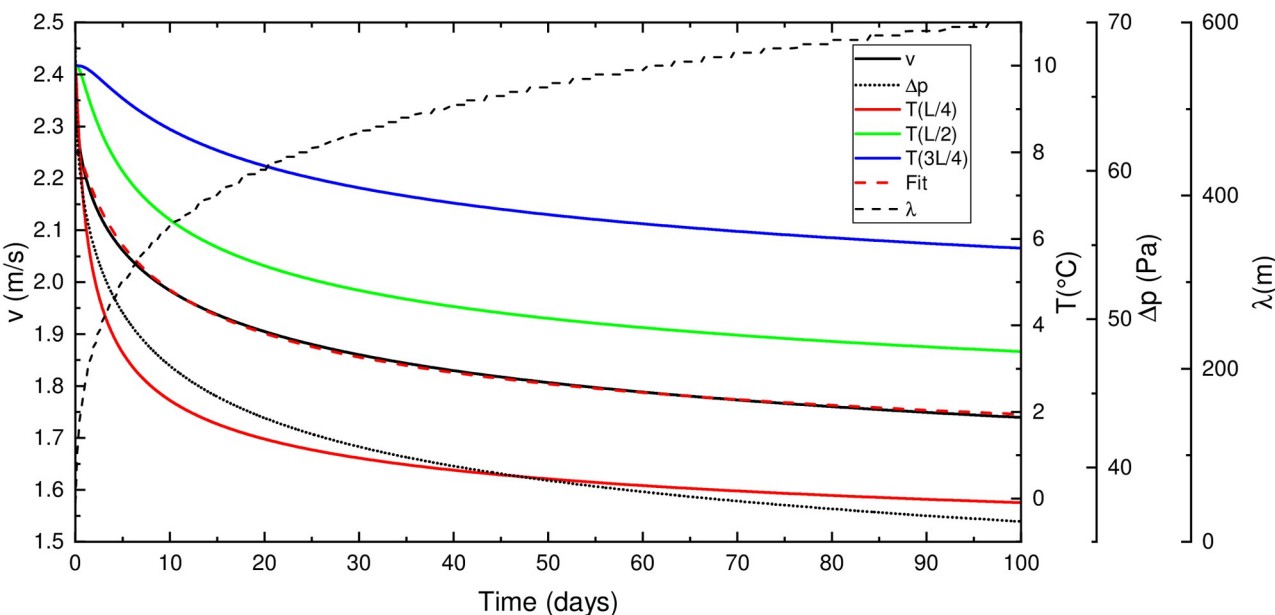

**Fig 6. Airflow velocity, driving pressure, relaxation length and temperature at three different locations in the tube for the case with the constant outside temperature at $T_{out} = -5$˚C.**

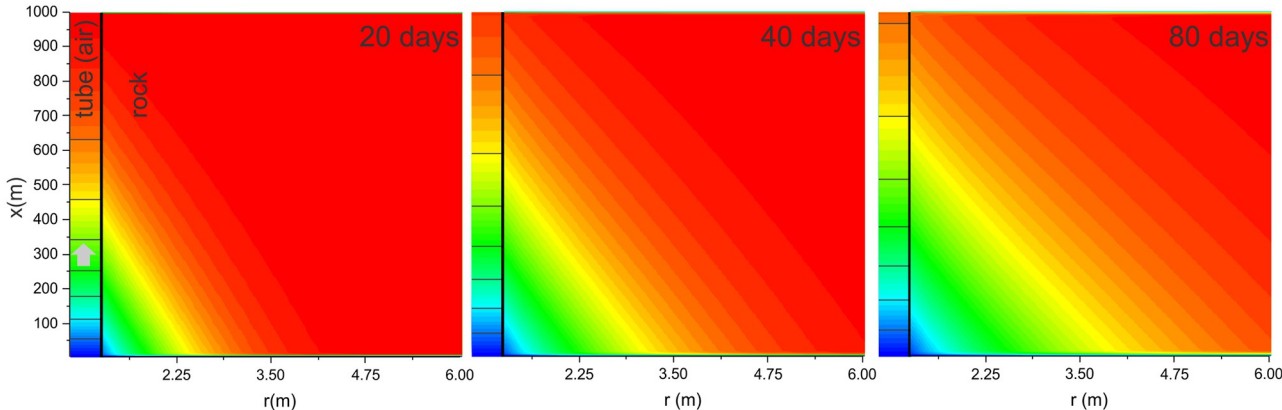

**Fig 7. Air temperature along the tube (left section, arrow indicating airflow direction) and the rock temperature in near tube region at 20, 40 and 80 days.**

Initially, the air along the entire tube is at $T_{in}$ resulting in maximal driving pressure and airflow velocity. The initial convective relaxation length is short, about 34 m, as predicted by Eq 8. As the conduction in the rock is limiting, so the thermal gradient builds up and the wall cools down. The penetration length, therefore, increases in time and reduces the density contrast and airflow. The increase in relaxation length and decrease in airflow velocity is a function of the square root of time. The red dashed curve in Fig 6 shows an almost perfect exponential fit to airflow velocity of the form $v(t) = v_0(1 - Ae^{-B/\sqrt{t}})$.

In a second scenario, the outside temperature is periodic of the form $T_{out}(t) = 10 + 15\sin(2\pi t/\tau)$, where $\tau$ is one year. Massif temperature is $T_{in} = 10°C$, as in the standard case.

Fig 8 shows the results. As expected, there is an updraft in the cold ($v > 0$) and a downdraft ($v < 0$) in the warm period. However, the maximal airflow velocities precede the maximal and

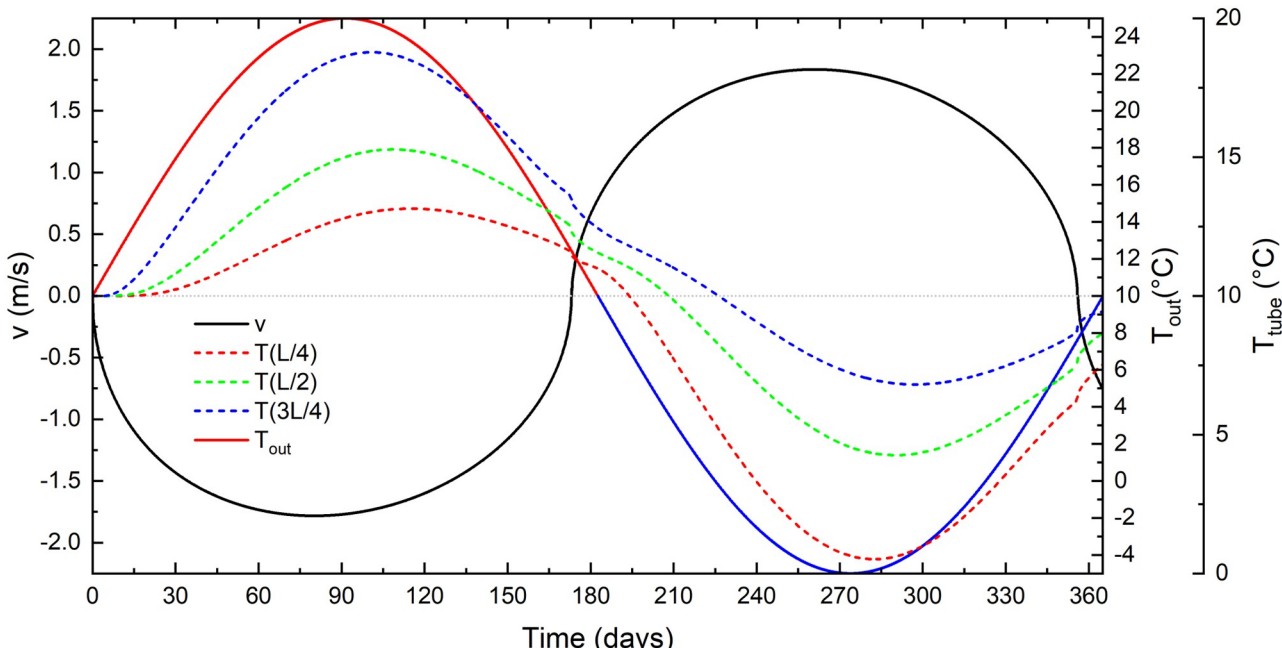

**Fig 8. Outside temperature (blue/red), airflow velocity (black) and air temperatures (dashed) at *L*/4, *L*/2 and 3*L*/2 for the standard case with periodic outside temperature.**

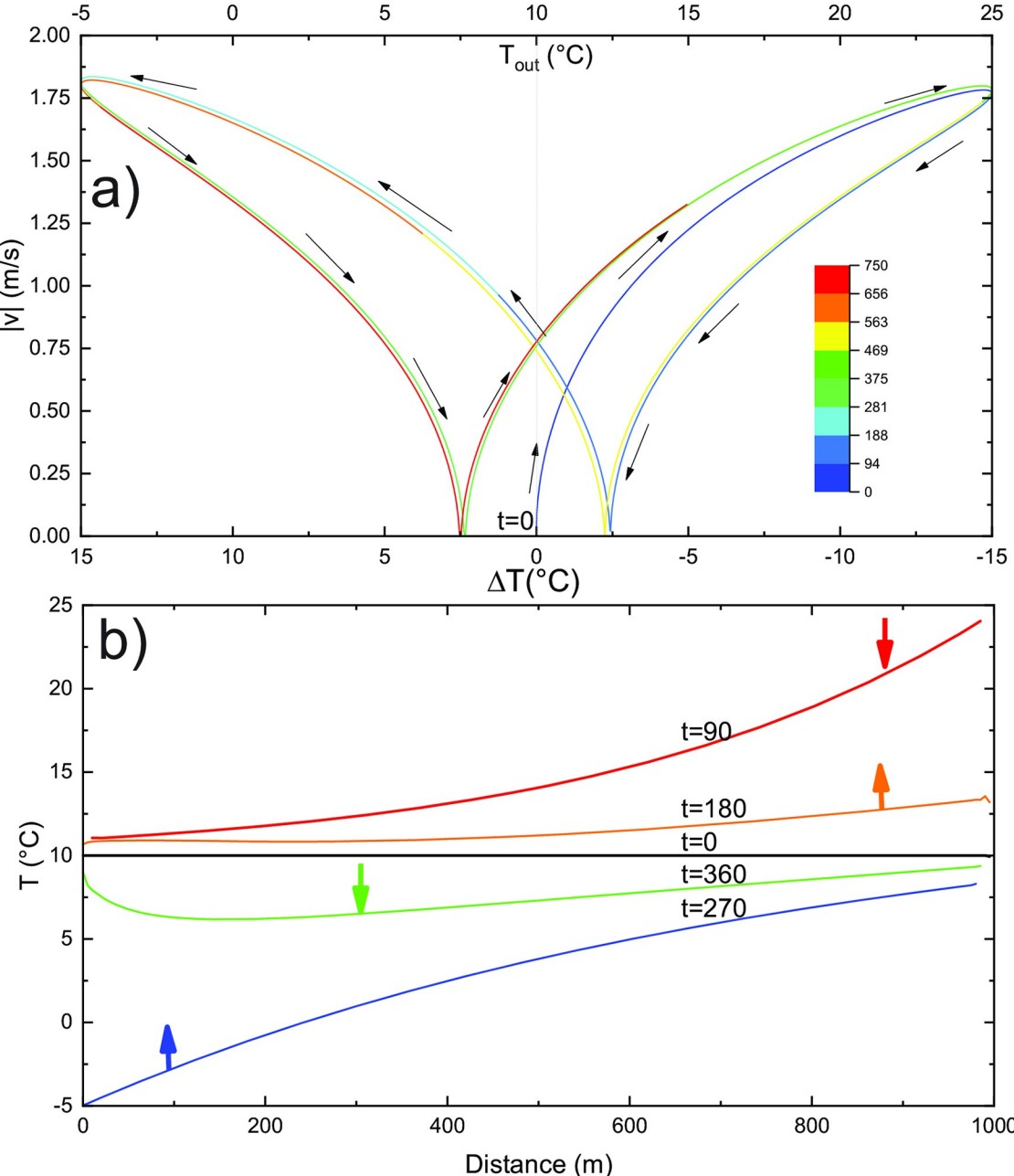

**Fig 9.** a) Absolute airflow velocity as a function of Δ*T* for the standard case with an annual temperature period. The upper axis shows the outside temperature. The arrows and curve colours indicate the progress in time. b) Temperature profile along the tube at four time steps for the basic case with an annual temperature cycle. Values denote the number of days from the start.

minimal external temperature due to conduction in rock and thermal inertia of the massif: heated walls and surrounding rock mass in the warm period keep the air in the tube warm during cooling of the outside air, so that the updraft begins before the outside air cools down to the temperature of the massif. The opposite happens in the transition from cold to warm period.

This is also demonstrated in Fig 9a, which shows an absolute airflow velocity as a function of Δ*T*. Here, the total time of simulation presented on the graph is 750 days. The colour of the

curve presents simulation days as shown by the bar code. Arrows indicate the progress of time. After the initial period (blue segment of the curve), the curve $|v|(T)$ follows an almost stable hysteresis curve with no airflow at $\Delta T = \pm 2.5°$C.

The hysteresis is an expected result of the massif's thermal inertia. The $|v|(T)$ curve still roughly follows a square-root relation between the outside temperature and the airflow velocity. The time lag between the phase of the outside temperature and the temperatures in the tube increases with distance from the inflow.

The points closer to the lower entrance are on average colder than the massif and the average outside temperature ($\bar{T}(L/4) = 8°$C), the points close to the upper entrance are on average warmer ($\bar{T}(3L/4) = 12°$C) while at midpoint $\bar{T}(L/2) \approx 10°$C.

Fig 9b shows the temperature profiles along the tube at four different time steps. Arrows at the curves indicate updraft or downdraft. As expected the temperature approaches that of the massif. In a pure convective case, the temperature at 180 days and 360 days would be constant and equal to $T_{in}$. This is not the case when heat conduction in the rock is considered; the thermal inertia of the system keeps the temperature along the tube above $T_{in}$ at 180 days and below $T_{in}$ at 360 days.

### Relation between airflow pattern and passage geometry

As discussed above, the changes in slope angle or cross-section may cause updraft or downdraft to be preferential. To check this, we apply the model to a simple L-case scenario and scenarios with a reduced cross-section at one of the entrances.

**Nonuniform longitudinal profile.** First, we take an L-shaped tube (Outline 1, see inset in Fig 10b), where 3/4 of the tube is horizontal at $z_1$ and the last 1/4 connects to the $E_2$ at the level, 100 m above $z_1$. Such situations of horizontal passages connected to the surface by a steep shaft are common in nature (see Fig 3). In a second scenario, we take a Γ-shaped tube (Outline 2) where the horizontal part is at the level $z_2$. All other parameters are as in the standard scenario.

The results shown in Fig 10 show that the airflow velocity in the dominant season (cold for L-shape and warm for Γ-shape) is up to three times higher for the same $|\Delta T|$ as in the non-dominant season. In both cases during the dominant season, airflow enters the massif at the entrance which connects to the horizontal section, so that the air in the vertical section is close to equilibrium with the massif. Due to the thermal inertia of the massif, the airflow reversals are shifted from $T_{in}$: for Outline 1, the wall in the vertical section is heated during downdraft in the warm season, which keeps the air in the tube warmer and less dense during the decrease of $T_{out}$, and the airflow reversal occurs already at about 15°C. Similarly, for Outline 2, where the vertical part is cooled effectively in the cold season, the airflow reversal to downdraft occurs at 5°C. The opposite reversals (updraft to downdraft for Outline 1 and downdraft to updraft in Outline 2) is closer to $\Delta T = 0$ because the air temperature in the vertical section is closer to equilibrium during the dominant airflow seasons. In Fig 10 two annual cycles are presented with almost stable hysteresis loops.

**Change of cross-section.** In the following scenarios, the diameter in the vicinity of one of the entrances is smaller. The results are presented in Figs 11 and 12. In the first case, (Cross 1) the diameter of the first 100 m long segment near the lower entrance is set to 0.5 m (the rest of the tube has ($D = 2$ m), in the second case (Cross 2) we do the same reduction for a 100 m long segment near the upper entrance. As expected from the reasoning given above, the change makes a notable asymmetry between the seasons: when reduction is at the upper entrance, the downdraft is preferential and vice versa; or simply the wind direction which first encounters the section with diameter reduction is preferential. The temperature at the mid-point $T(L/2)$ is close to the $T_{in}$ during the period when air enters through a small diameter section. Note a

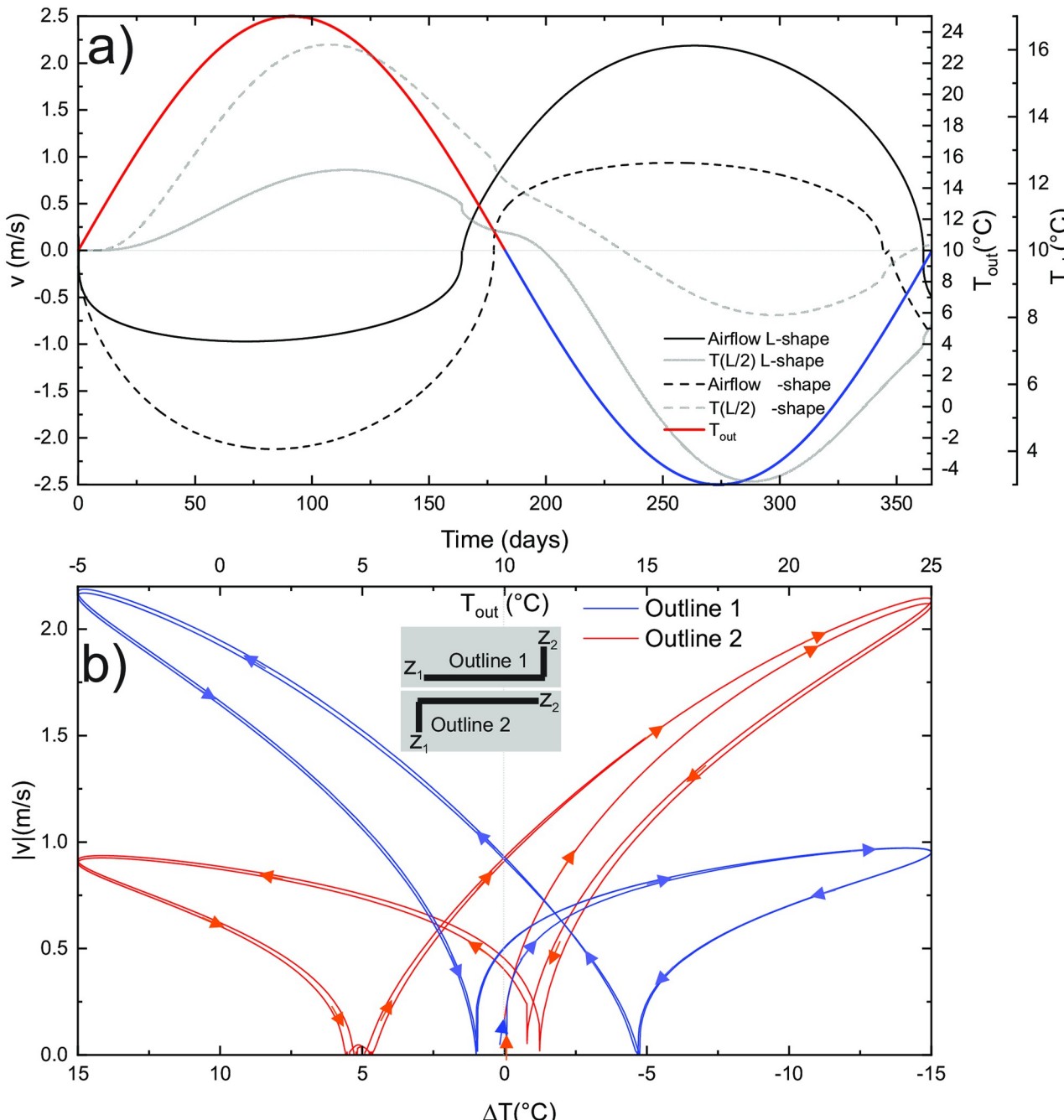

**Fig 10.** a) Airflow velocities, external temperature and temperature at $L/2$ during the annual cycle for the two nonuniform outlines sketched in the insert in Figure b. b) Airflow pattern $|v(\Delta T)|$ for the two different outlines. Arrows indicate the progress of time. The total simulation time is 750 days.

smaller hysteresis in Fig 12 during the dominant season, which is a consequence of a shorter relaxation length.

**V-shaped longitudinal profile.** It is common in nature that deep caves have multiple entrances at high elevations (such as in high karst plateaus), which may join in the depth. The basic building block for understanding ventilation in such systems is a V-shaped passage,

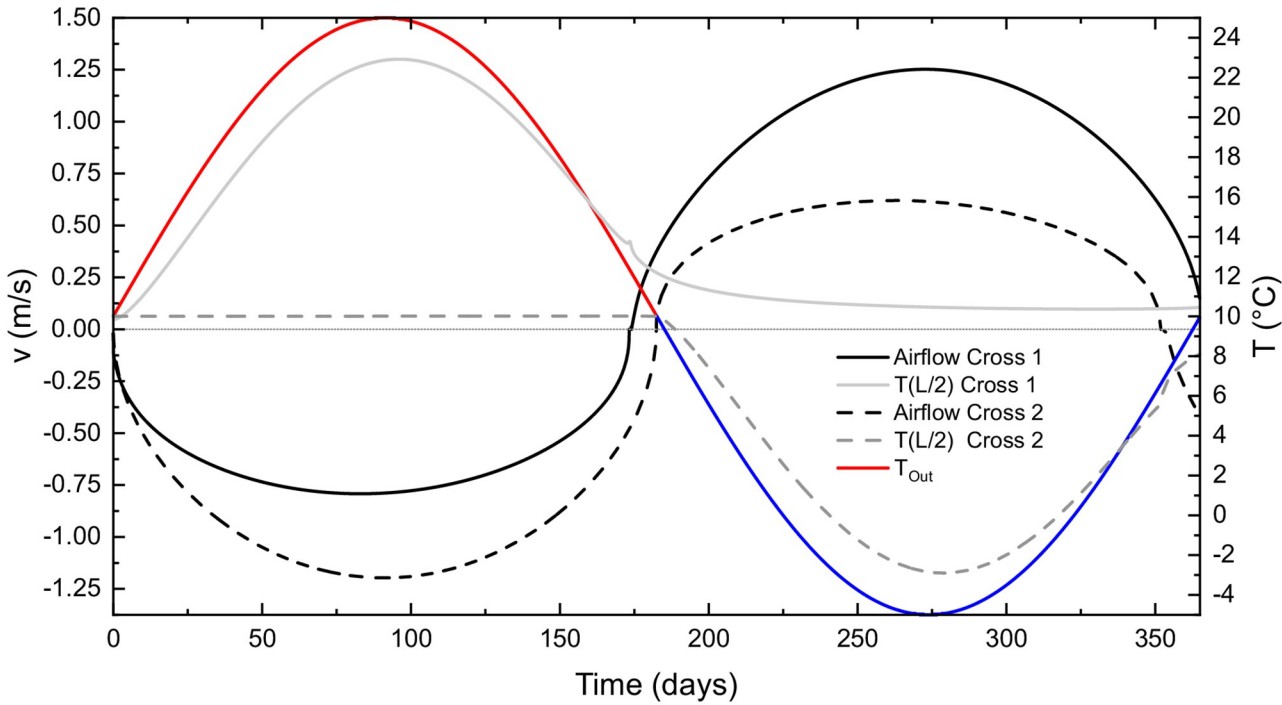

**Fig 11. Airflow velocities, external temperature and temperature at $L/2$ during annual cycle for the two cases with change in pipe diameter (see insert in Fig 12).**

shown in Fig 13a. In this case, it is not the disequilibrium between the internal and external air column which drives the airflow, but the density difference between both limbs of the tube.

Fig 13 shows the time evolution of airflow velocity for the V-shape system with constant $T_{out} = -5°C$. To induce the flow, the elevation at the left entrance is slightly higher, $\delta z = 0.03$ m (Fig 13a). All other parameters are as in the standard case.

Fig 13a shows the relaxation length and airflow velocity for the first 10 min. Initially, the temperature in both limbs is equal to $T_{in}$. However, even an infinitesimally small intrusion of cold air, in this case, triggered by a 3 cm higher positioned right entrance, breaks the

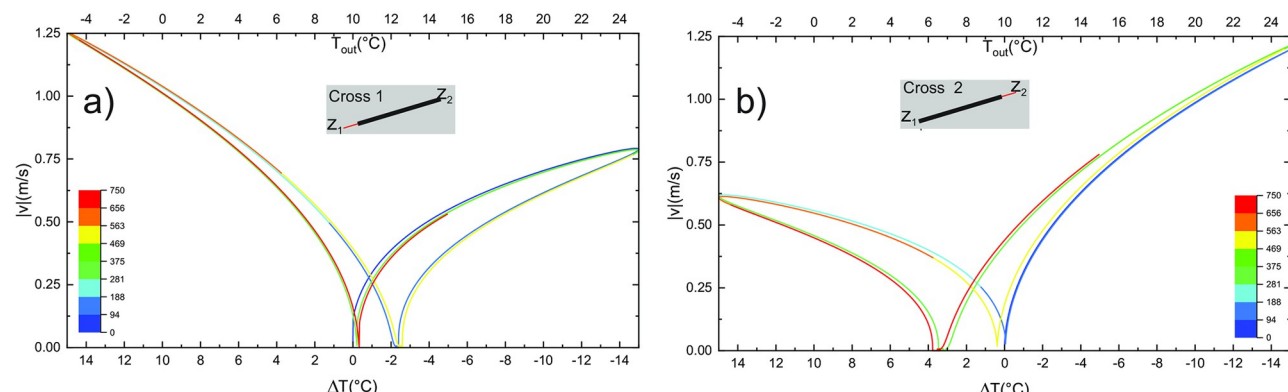

**Fig 12. Airflow pattern $|v(\Delta T)|$ for scenarios with change in cross-sections as shown in the insert.** Colours indicate the progress in time as given in the bar codes.

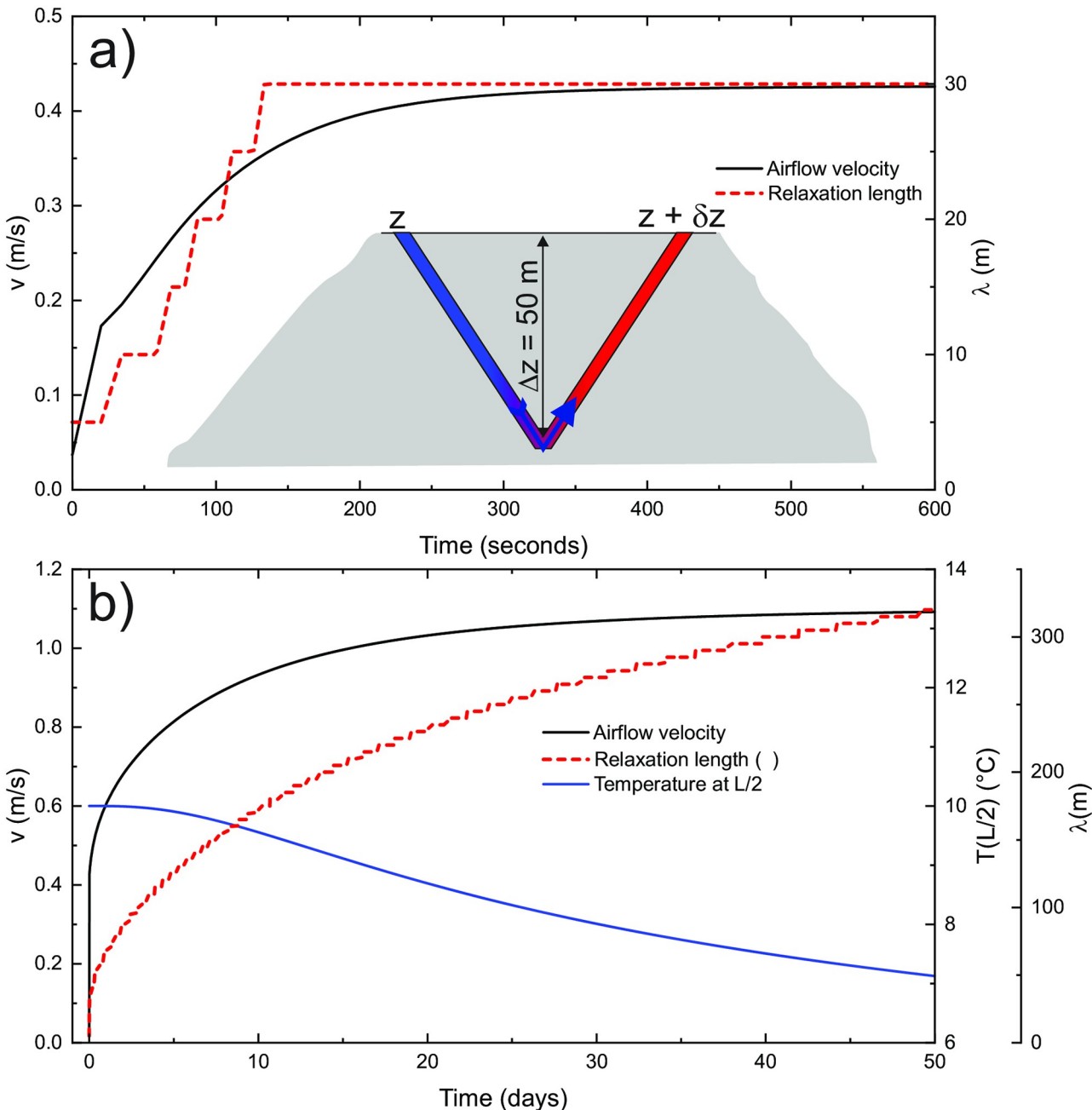

**Fig 13.** a) Initial evolution of the airflow velocity and relaxation length for a V-shape case with constant external temperature ($T_{out}$ = −5˚C). The insert shows a geometry of the system, $L$ = 1 km, $D$ = 2 m, $\delta z$ = 0.03 m. b) Long-term evolution of airflow velocity, relaxation length and temperature at the knickpoint, $T(L/2)$.

equilibrium. The feedback mechanism between the density disequilibrium and airflow is initiated, which causes a fast initial rise in airflow velocity. The rise is dampened by increasing flow resistance. After 10 min the system is in a "convective equilibrium", but the cooling of the cave walls causes a further increase of the relaxation length and the airflow velocity as shown in Fig 13b. The rise is limited by the penetration of cold air into the right limb; for a hypothetical case where $\lambda >> L$, the system would be at rest.

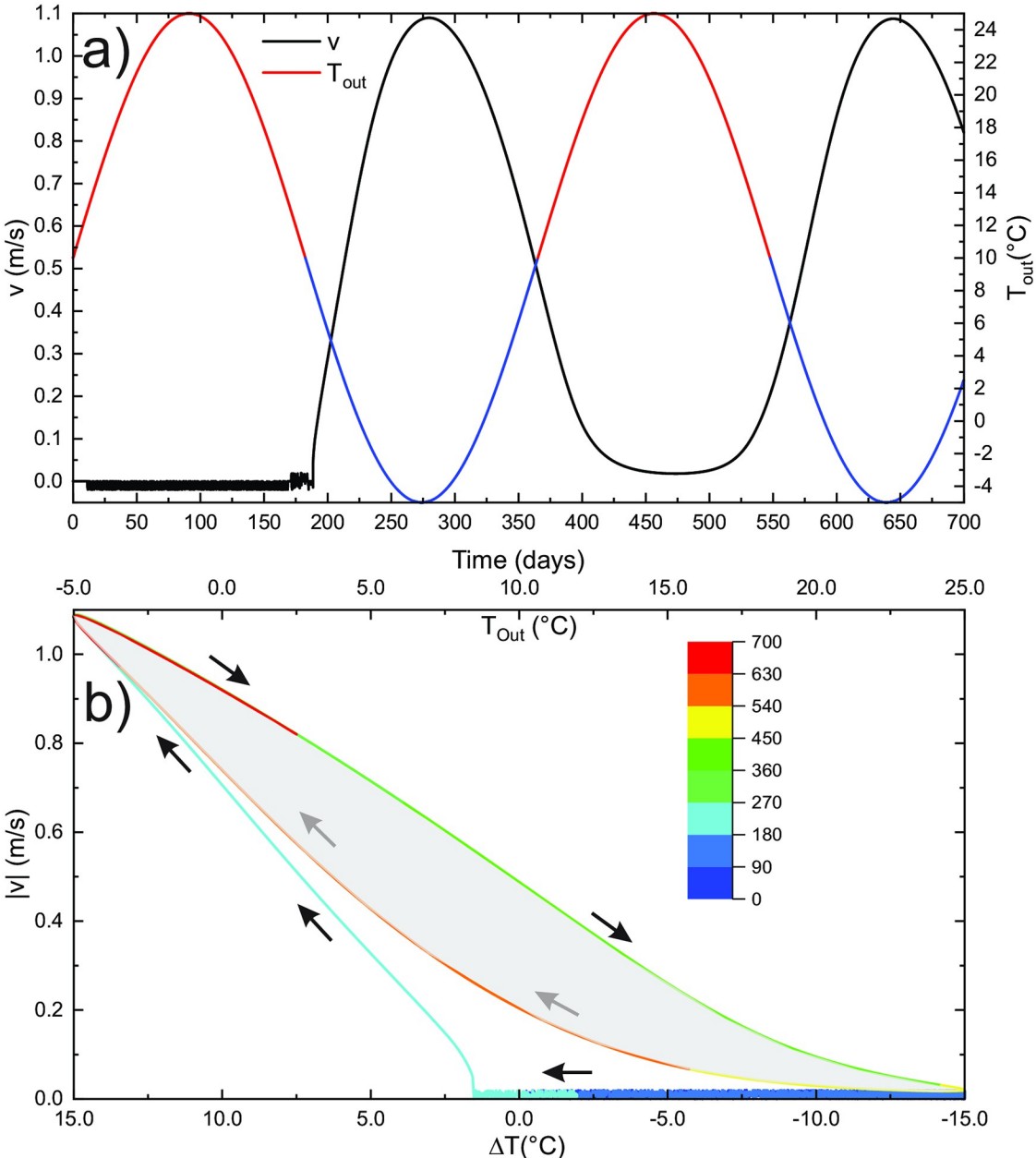

**Fig 14.** a) Airflow velocity and external temperature in a V-profile for 750 days of the standard periodic cycle. b) Airflow pattern, $|v (\Delta T)|$, follows a hysteresis loop encircling the grey zone.

Lismonde [27] introduced the concept of instability in the V-shaped systems. Latter Fainmon and Lang [20] also noted the feedback mechanism between intruding cold air and airflow, and related it to the nonlinear relation between air density and temperature.

During warm periods $\Delta T < 0$, the system is stable. The intrusion of warm outside air at one side would make the other side heavier and the warm air would be pushed back out. This negative feedback was also mentioned by Fainmon and Lang [20].

Fig 14 shows the airflow and outside temperature in V- shape profile for a standard cycle for 700 days. Initially, the system is at rest at $\Delta T > 0$. The airflow is triggered at $\Delta T \approx 2°C$ and

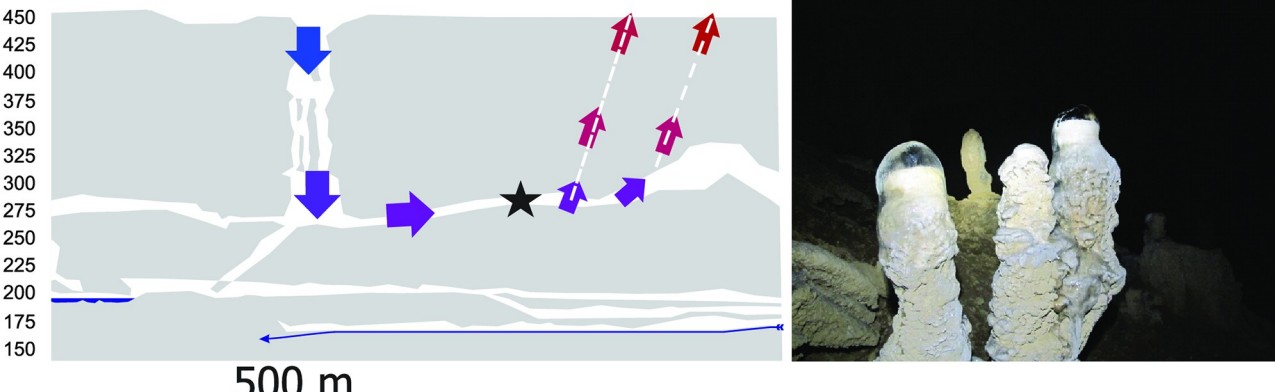

**Fig 15.** Left: A simplified profile of part of the Kačna Cave (Slovenia). Arrows show winter airflow direction and air temperature on a red (warm) to blue (cold) colour scheme. Dashed lines indicate unexplored airflow pathways. Right: photograph of ice formation on the stalagmites taken during cold period in February 2012. The approximate position of stalagmites is indicated by a black star in the profile.

airflow velocity increases with decreasing outside temperature. However, the airflow velocity remains positive throughout the warm period due to the thermal inertia of the system (Fig 14b). If the standard cycle continues, the curve $|v|(\Delta T)$ stays in a hysteresis loop presenting the boundary of the grey surface in Fig 14b.

V-system is a cold trap, with an inflow of cold air during the cold period and almost no airflow during the warm period, which makes the settings ideal for the formation of ice caves [21]. Even more, such systems may (and often do) have large passages with long relaxation lengths on one side and small passages with short relaxation lengths on the other side. This makes the system even more effective as it allows penetration of cold air to the knickpoint.

The same reasoning with inverse results are obtained for the inverse case of $\Lambda$-shape, when the system is unstable in a warm period and stable in cold period.

The above mechanism is present in Kačna Cave, a large cave system in Classical Karst in Slo-venia (Fig 15) There, the main entrance is 180 m deep shaft with diameters above 20 m. The cave continues along large passages with no other known exit to the surface. In summer there is almost no airflow in the absence of external winds. In winter, however, the cold air pene-trates along the entrance shaft deep into the cave and must exit to the surface along unknown airflow pathways. The cold air cools down the rock walls and other formations so that the drip water from the ceiling freezes on the surface of stalagmites and forms ice several hundred meters away from the entrance shaft (Fig 15).

Most V-shaped caves are not perfect but have considerable altitude differences between the entrances. In these cases the outside air column has to be added to the pressure of the shorter limb to obtain the difference; the analytical expression given later can be used to assess the driving pressure for a general case.

## The presence of ice

Changes in ventilation patterns and microclimate in caves can be caused by several mechanisms. Caves may accumulate ice or snow during cold periods, when snow may slide down an inclined passage and/or ice may form due to freezing of the seepage water. In certain settings (especially cold traps), snow and ice accumulated during the cold period may remain in the cave throughout the year, resulting in an ice cave [21, 28]. It was shown here that certain conduit geometries result in cave air temperatures below the average surface temperature, which

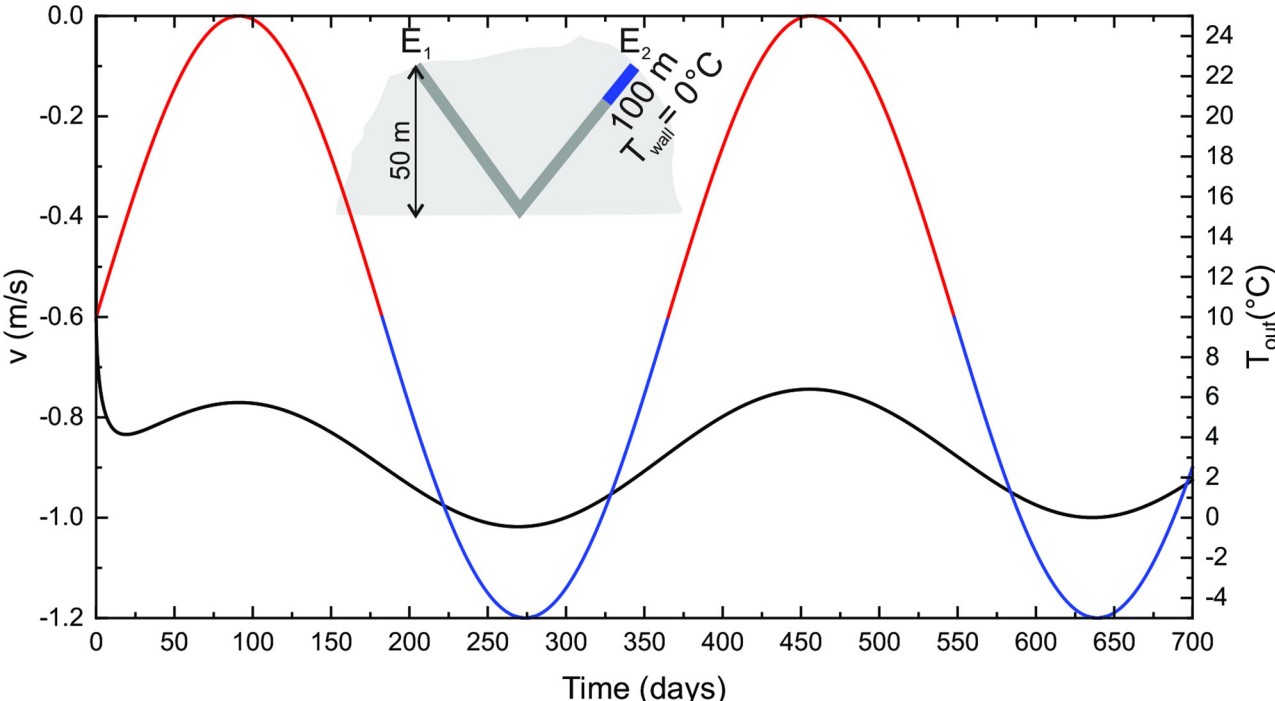

**Fig 16. Airflow velocity and outside temperature for a V-shape system, where wall temperature along the section marked blue is set to 0°C.**

can result in perennial ice formations in areas with average external temperatures well above 0°C.

The accumulation of snow and the formation of ice is beyond the scope of this work. We limit the discussion on how the existence of ice may change the ventilation pattern. To do that we set the wall temperature in a selected segment of the tube to 0°C.

Fig 16 shows the airflow velocities for the same V-shape setting as presented in a previous section, with the wall temperature along the 100 m long segment at the right entrance ($E_2$) set to 0°C. The airflow has permanent direction $E_2 \rightarrow E_1$. The velocity is higher in the winter but remains high also through the warm season. We have observed a similar airflow pattern in one of the deep caves on high Dinaric mountain Snežnik in Slovenia. The cave is more than 600 m deep, with the entrance almost at the top of a plateau. Snow presence in the entrance shaft persists deep into the warm season. The downdraft has been observed in the cave in all seasons to the deepest explored point. Assuming that the cave at some point connects to another cave with an entrance at the plateau, the observed pattern could be explained by the same concept.

One can envisage many scenarios with ice present at different segments of the passages. However, to explore the role of ice in detail, ice accumulation and melting need to be included in the model.

## Conclusions

The airflow pattern in caves is a complex phenomenon governed by multiple factors. To understand it, we need to break it down into simple building blocks and concepts. Some of these are presented in this work.

It was shown that non-uniform passages show seasonal airflow asymmetry. While an L-shaped passage has stronger airflow during updraft in a cold period, the opposite is valid for an

inverse $\Gamma$-shaped passage. If a passage cross-section changes, the airflow direction with a shorter penetration length is dominant. In a V-shaped passage with entrances at similar altitudes, a perturbation causes a feedback loop between airflow and penetration length. The system is however stable with no airflow during the warm season. The opposite applies to a $\Lambda$-shaped passage.

The results are based on approximations and assumptions that are not always valid. Variations in humidity or $CO_2$ concentrations may in the absence of temperature changes be the main driving force of airflow [19]. During the warm period, the production of $CO_2$ in the soil is higher, and also the cave air is generally more enriched with it, which gives an additional boost to the downdraft. When $T_{out} \approx T_{in}$ the variations of $CO_2$ may even be the main airflow driver. Evaporation/condensation processes do not only influence the density directly but also play role in the heat exchange. During updraft, the cold outside air warms up and dries along the passage. If the walls are wet, the walls and air may be cooled by evaporation, which extends the penetration length and diminishes the driving pressure. During downdraft in a warm period, the air is cooled by the massif, the relative humidity rises ad water may condense on the wall; the produced latent heat prolongs the penetration length resulting again in lower driving pressure. The inclusion of evaporation/condensation processes is among the first planned upgrades. This could also give an assessment of the speleogenetic role of air moisture (condensation corrosion) and the contribution of air moisture to the recharge of karst aquifers.

The discussion on ice caves is limited by an assumption of a stagnant ice section. The next step is to introduce the formation and melting of ice into the model and look for its long-term relation between outside climate, cave geometry and ice accumulation.

The geometry of the system is an idealisation; most of the caves have multiple passages and entrances at different altitudes. How airflow patterns evolve in more complex systems is a challenging question, which could be addressed with further development of the model.

However, even for simple settings presented here, the modelling results are only snapshots into a more general picture. To this extent, an analytical approximation for a driving pressure in a double slope passage for a convection-only regime is presented in S1 Appendix, which gives a more general relation between the basic parameters and driving pressure.

In this work, the comparison of the modelling results with field data is only qualitative. Quantitative fits are currently beyond the scope of but are possible with measurements in systems with well-constrained settings.

## Supporting information

**S1 Appendix. An a nalytical approximation for a driving pressure.** See the appendix for a derivation of the analytical approximation for a driving pressure in a general double-slope passage with an exponential temperature profile.
(PDF)

## Acknowledgments

This work benefited from discussions with Matt Covington (University of Fayetville, USA) and my PhD student Lovel Kukuljan. Data from Postojnska Jama were obtained from the cave meteorological stations, which wouldn't be working without the ingenious technical skills of Boštjan Grašič and Primož Mlakar. The work of the late Giovanni Badino, physicist, speleologist and one of the foremost researchers of cave climate inspired my interest in the topic.

## Author Contributions

**Conceptualization:** Franci Gabrovšek.

**Data curation:** Franci Gabrovšek.

**Funding acquisition:** Franci Gabrovšek.

**Investigation:** Franci Gabrovšek.

**Methodology:** Franci Gabrovšek.

**Project administration:** Franci Gabrovšek.

**Software:** Franci Gabrovšek.

**Visualization:** Franci Gabrovšek.

**Writing – original draft:** Franci Gabrovšek.

**Writing – review & editing:** Franci Gabrovšek.

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
