## [Decision Letter · Decision Letter 0]

13 Jan 2023

PONE-D-22-29357How do caves breath: the airflow patterns in karst undergroundPLOS ONE

Dear Dr. Gabrošsek,

Thank you for submitting your manuscript to PLOS ONE. After careful consideration, we feel that it has merit but does not fully meet PLOS ONE’s publication criteria as it currently stands. Therefore, we invite you to submit a revised version of the manuscript that addresses the points raised during the review process.

We look forward to receiving your revised manuscript.

Kind regards,

Yanping Yuan

Academic Editor

PLOS ONE

Journal Requirements:

"The work was supported by the Slovenian Research Agency (grant No. L6-9397)."

"FG was was funded by the Slovenian Research Agency as part of the Research projects L6-9397 and Research programme P6-0119."

"FG was was funded by the Slovenian Research Agency as part of the Research projects L6-9397 and Research programme P6-0119." 

6. We note that Figures 3a and 3d in your submission contain map images which may be copyrighted. All PLOS content is published under the Creative Commons Attribution License (CC BY 4.0), which means that the manuscript, images, and Supporting Information files will be freely available online, and any third party is permitted to access, download, copy, distribute, and use these materials in any way, even commercially, with proper attribution. For these reasons, we cannot publish previously copyrighted maps or satellite images created using proprietary data, such as Google software (Google Maps, Street View, and Earth). For more information, see our copyright guidelines: http://journals.plos.org/plosone/s/licenses-and-copyright.

  a. You may seek permission from the original copyright holder of Figures 3a and 3d to publish the content specifically under the CC BY 4.0 license.  

Reviewers' comments:

Reviewer's Responses to Questions

**Comments to the Author**

1. Is the manuscript technically sound, and do the data support the conclusions?

Reviewer #1: Yes

Reviewer #2: Yes

2. Has the statistical analysis been performed appropriately and rigorously? 

Reviewer #1: N/A

Reviewer #2: Yes

3. Have the authors made all data underlying the findings in their manuscript fully available?

Reviewer #1: Yes

Reviewer #2: Yes

4. Is the manuscript presented in an intelligible fashion and written in standard English?

Reviewer #1: Yes

Reviewer #2: Yes

5. Review Comments to the Author

Reviewer #1: The article describes the ventilation in caves with 2 entries, commonly called "chimney-effect", where the direction and velocity of the airflow is proportional to the internal/external temperature difference and to the altitude difference between both. This classical aspect is described in the first part. One would expect a perfect seasonal symmetry. Based on long measurements of these parameters in natural sites, the author shows on the contrary a clear seasonal asymmetry. This very interesting new observation is formalized in the second part by a conceptual model (whose physics I am not able to validate), on which a numerical model is then built to examine the role of the variations of vertical profile of passages, of width changes, etc. The results are discussed, showing in particular the orders of magnitude of the airflow variations according to the considered cases. The paper concludes by examining a V-shaped duct with two inlets at the same altitude, where the driving pressure is theoretically zero in summer, but where the winter instability is prolonged beyond the cold season due to the thermal inertia of the rock. These results confirm and extend those of Lismonde 2001 (https://doi.org/10.3406/karst.2001.2468). Again, the shape of the inlets, where the presence of a cold mass (snow or ice), significantly disrupts the theoretical scheme.

This paper presents considerable advances in the understanding of air flows previously described as "simple", advances made possible by numerical tools. Future implications are expected, on the conservation of now disappearing unground ice, subsurface condensation water, distribution of carbonate dissolution in the vadose part, etc. As such, it deserves to be published in this type of Journal.

I have no major substantive remarks, it is clearly written in both style and approach, and it can be published "as it is", considering the following very minor corrections:

- Paragraph "Driving pressure of the chimney effect..." (no line numbering here): Starting from the inlet E1 (not E2!) at the elevation z1 and pressure p1...

- 124 : author mentions "most of our observation sites...", whereas only 2 sites are documented before. Only at the final end of the paper, he discusses other sites. Be clearer, and make the difference between qualitative field observations and studied sites with quantitative records

- 293: In following scenarios the diameter in the vicinity...

- 384: The results are based on approximations and assumptions that are (in < remove) not always valid

Philippe Audra

Reviewer #2: More and more studies recognize the connection and joint control of air and water circulation on the processes of karstification that create and continuously modify karst systems. This study represents a significant step in the numerical modeling of air circulation by the chimney effect mechanism, which seems to be the most significant driver of air circulation, especially in the shallower parts of the karst subsurface. Modeling describes the features of seasonal air circulation for several typical morphological forms of caves. The presented results quantitatively show the features of the circulation, and are particularly useful for interpreting the results of the monitoring performed in-situ in the caves. Therefore, I highly recommend the manuscript for publication with a few minor suggestions.

Specific comments:

Lines 103-123: It seems to me that the described example is mostly based on the study published by Kukuljan et al. 2021, which is not noted in this section (otherwise monitoring setup should be described in more detail).

Fig. 13, Lines 313-322: According to the figure (and description within the text) it seems that in the case of colder outside air (-5°C), the air circulation in the cave is directed from the higher entrance (z+δz) to the lower one (downdraft). This is in contrast to the previous examples (updraft during the period of colder outside air), and also contrary to the described example of the "Kačna" Cave (Figure 15), where during the cold season the cold air enters through the lower (main) entrance of the cave. So, it looks that there is a mistake on the Figure 13 regarding airflow direction?

Line 323: Luetcher et al (2008) also noted cold trap behavior of u-shaped cave.

Figure 15: Short explanation for the varying colors of the arrows indicating airflow direction could be added.

Lines 368-380, Figure 16: Example of "Snežnik" Cave is interesting but slightly confusing: if the mentioned entrance is at the highest point of the massif, the formation of ice in it should indicate a downdraft during the cold season, which contradicts the previous considerations?

Also, the model shown in Figure 16 includes two inlets approximately at the same height, while the Snežnik example probably has a "snow covered" inlet significantly above the other one? Not sure if the model would show a similar results for a configuration more suitable for Snežnik example.

There is a recently published study of monitoring results from a very deep cave with extensive ice deposits (Velebit Mt., Dinaric Karst), which interprets the functioning of the cave as a cold-trap that causes a negative temperature anomaly within the karst massif (in accordance to your results and considerations).

Conclusion section: I propose to additionally include brief conclusions related to the influence of object morphology (L-shaped, V-shaped, varying diameter) on seasonal airflow patterns.

6. PLOS authors have the option to publish the peer review history of their article (what does this mean?). If published, this will include your full peer review and any attached files.

Reviewer #1: **Yes: **Philippe AUDRA

Reviewer #2: No

---

## [Author Response · Author response to Decision Letter 0]

14 Feb 2023

I have responded to all reviewer's and editor's comment in cover letter and response to reviewers. I have uploaded both files. I am copy-pasting these responses also to this form:

Answer: I have used a LaTeX template downloaded from PLOS site.

"The work was supported by the Slovenian Research Agency (grant No. L6-9397)."

"FG was was funded by the Slovenian Research Agency as part of the Research projects L6-9397 and Research programme P6-0119."

Answer: I have removed the statement from the Acnowledgement and changed the funding statement. Below I am also adding the 

The funding statement now reads (please change the online form accordingly):

This work was was funded by the Slovenian Research Agency as part of the research projects L6-9397 and J7-4630, and the research programme P6-0119.

"FG was was funded by the Slovenian Research Agency as part of the Research projects L6-9397 and Research programme P6-0119." Please state what role the funders took in the study. If the funders had no role, please state: "The funders had no role in study design, data collection and analysis, decision to publish, or preparation of the manuscript." 

The funder statement now reads (please change the online form accordingly):

The funder had no role in study design, data collection and analysis, decision to publish, or preparation of the manuscript.

Has been done.

Answer: the data is available at our institute’s data depository. Link to the data:

https://cloud.izrk.zrc-sazu.si/index.php/s/s2oTxfqDAADfJJd

If there are any further requirements, I’ll be happy to follow them.

6. We note that Figures 3a and 3d in your submission contain map images which may be copyrighted. All PLOS content is published under the Creative Commons Attribution License (CC BY 4.0), which means that the manuscript, images, and Supporting Information files will be freely available online, and any third party is permitted to access, download, copy, distribute, and use these materials in any way, even commercially, with proper attribution. For these reasons, we cannot publish previously copyrighted maps or satellite images created using proprietary data, such as Google software (Google Maps, Street View, and Earth). For more information, see our copyright guidelines: http://journals.plos.org/plosone/s/licenses-and-copyright.

Answer: Figures 3a and 3d are adopted from the work of Kukuljan et al (2021), where I was a corresponding authors. Both works are published under CC BY 4.0. https://doi.org/10.5038/1827-806X.50.3.2392, https://doi.org/10.1007/s00704-021-03722-w

I hope this works. If not, I will change/remove the maps and forward the reader to the cited publications.

Answer: I hope I have done it as required.

Answer: I have reviewed the reference list. The reference style was included in the LaTex package available at PLOS site. I have added one new reference (Lismonde, 2001), number 27 in the reference list, as suggested by the Reviewer 1.

Response to reviewers

I sincerely thank to both reviewers for recognising the importance of this work and suggestions for improvement. Below find a list of comments and my answers and reactions to them. All line number given correspond to numbering in the file with marked changes (Gabrovsek-plos-tracked.pdf)

REVIEWER 1:

Comment: The reviewer reminds on the work of Lismonde (2001) on V-shaped passage. I have (with my limited French and electronic translation) read the paper, which recognises the instability in the V-shaped passage. In my work I elaborate the idea into a numerical result. 

Response: I have now included citation and I mention the findings in the text. Line 353 in the tracked version, Reference 27.

Comment: Paragraph "Driving pressure of the chimney effect..." (no line numbering here): Starting from the inlet E1 (not E2!) at the elevation z1 and pressure p1...

Response: Corrected. Line 101 in tracked version.

Comment: 124 : author mentions "most of our observation sites...", whereas only 2 sites are documented before. Only at the final end of the paper, he discusses other sites. Be clearer, and make the difference between qualitative field observations and studied sites with quantitative records.

Response: Corrected: Line 145 to 148 in tracked version.

Comment: 293: In following scenarios the diameter in the vicinity...

Response: Corrected. Line 322 in tracked version.

Comment: 384: The results are based on approximations and assumptions that are (in < remove) not always valid

Response: Corrected. Line 423 in tracked version.

REVIEWER #2: 

Comment: Lines 103-123: It seems to me that the described example is mostly based on the study published by Kukuljan et al. 2021, which is not noted in this section (otherwise monitoring setup should be described in more detail).

Response: I added the text stating that the measurement system is described in Kukuljan (2021): Lines 142-144 in tracked version.

Comment: Fig. 13, Lines 313-322: According to the figure (and description within the text) it seems that in the case of colder outside air (-5°C), the air circulation in the cave is directed from the higher entrance (z+δz) to the lower one (downdraft). This is in contrast to the previous examples (updraft during the period of colder outside air), and also contrary to the described example of the "Kačna" Cave (Figure 15), where during the cold season the cold air enters through the lower (main) entrance of the cave. So, it looks that there is a mistake on the Figure 13 regarding airflow direction?

Response: In a V-shaped passage it is the imbalance (or density difference) between the both limbs of the passage that triggers the airflow. To trigger the initial airflow, which drives the system into a feed-back loop, here a minimal elevation difference δz between both entrances was used. However, as the reviewer recognised this would cause flow from lower to higher entrance, which is the case. So the direction on Figure 13a is wrongly marked and has been corrected.

The topography above Kačna jama is drawn schematically. The actual airways where air returns to the surface are not really known; but they must exist. Also not known is the position of oulets (blow holes) at the surface. The initial imbalance may ibe caused by the small altitude difference or by other pertrubations, such as surface winds. I have redrawn Figure 15 and changed the caption to avoid confusion.

See Fig 15 and caption between lines 386 and 387 in tracked version.

Comment: Line 323: Luetcher et al (2008) also noted cold trap behavior of u-shaped cave.

Response: Citation to Luetcher et al. (2008) is added in line 392 (tracked version), where importnce of cold traps for the formation of ice caves is mentioned.

Comment: Figure 15: Short explanation for the varying colors of the arrows indicating airflow direction could be added.

Response: Added. See caption of Fig 15, between lines 386 and 387 in tracked version.

Comment: Lines 368-380, Figure 16: Example of "Snežnik" Cave is interesting but slightly confusing: if the mentioned entrance is at the highest point of the massif, the formation of ice in it should indicate a downdraft during the cold season, which contradicts the previous considerations? Also, the model shown in Figure 16 includes two inlets approximately at the same height, while the Snežnik example probably has a "snow covered" inlet significantly above the other one? Not sure if the model would show a similar results for a configuration more suitable for Snežnik example.

There is a recently published study of monitoring results from a very deep cave with extensive ice deposits (Velebit Mt., Dinaric Karst), which interprets the functioning of the cave as a cold-trap that causes a negative temperature anomaly within the karst massif (in accordance to your results and considerations).

Response: Maybe giving Snežnik cave as an example might be a bit of a speculation. However, we know only one entrance leading to 630 m deep system of shafts with. We have observed dowdraft in the cave in all seasons and the presence of ice through most of the seasons between depths of 100 m and 200 m. The outlet of the air could be higher, but according to local topography it is hard to imagine that it is significantly higher; maybe 100 m, which is, taking into account that the downdraft is still present at the depth of 600 m, not very significant. However, idea that the presence of ice in the known limb keeps the air density in the limb high and sustaind dowdraft in all seasons.I am familiar with research on Velebit mountain. I am not very sure which publication the reviewer has in mind. However, I believe that the model could be used to interprete many »unusual« situations.

Comment: Conclusion section: I propose to additionally include brief conclusions related to the influence of object morphology (L-shaped, V-shaped, varying diameter) on seasonal airflow patterns.

Response: Done, see lines 416-421 in tracked version.

---

## [Decision Letter · Decision Letter 1]

16 Mar 2023

How do caves breathe: the airflow patterns in karst underground

PONE-D-22-29357R1

Dear Dr. Gabrovšek,

We’re pleased to inform you that your manuscript has been judged scientifically suitable for publication and will be formally accepted for publication once it meets all outstanding technical requirements.

Kind regards,

Yanping Yuan

Academic Editor

PLOS ONE

Additional Editor Comments (optional):

Reviewers' comments:

Reviewer's Responses to Questions

**Comments to the Author**

1. If the authors have adequately addressed your comments raised in a previous round of review and you feel that this manuscript is now acceptable for publication, you may indicate that here to bypass the “Comments to the Author” section, enter your conflict of interest statement in the “Confidential to Editor” section, and submit your "Accept" recommendation.

Reviewer #2: All comments have been addressed

2. Is the manuscript technically sound, and do the data support the conclusions?

Reviewer #2: Yes

3. Has the statistical analysis been performed appropriately and rigorously? 

Reviewer #2: Yes

4. Have the authors made all data underlying the findings in their manuscript fully available?

Reviewer #2: Yes

5. Is the manuscript presented in an intelligible fashion and written in standard English?

Reviewer #2: Yes

6. Review Comments to the Author

Reviewer #2: (No Response)

7. PLOS authors have the option to publish the peer review history of their article (what does this mean?). If published, this will include your full peer review and any attached files.

Reviewer #2: **Yes: **Andrej Stroj

---

## [Editor Report · Acceptance letter]

24 Mar 2023

PONE-D-22-29357R1 

How do caves breathe: the airflow patterns in karst underground 

Dear Dr. Gabrovšek:

I'm pleased to inform you that your manuscript has been deemed suitable for publication in PLOS ONE. Congratulations! Your manuscript is now with our production department. 

Kind regards, 

on behalf of

Prof. Yanping Yuan 

Academic Editor

PLOS ONE